# Globally Convergent Variational Inference

**Declan McNamara**                                        DECLAN@UMICH.EDU
**Jackson Loper**                                          JALOPER@UMICH.EDU
**Jeffrey Regier**                                         REGIER@UMICH.EDU
*Department of Statistics, University of Michigan*

## Abstract

In variational inference (VI), an approximation of the posterior distribution is selected from a family of distributions through numerical optimization. With the most common variational objective function, known as the evidence lower bound (ELBO), only convergence to a *local* optimum can be guaranteed. In this work, we instead establish the *global* convergence of a particular VI method. This VI method, which may be considered an instance of neural posterior estimation (NPE), minimizes an expectation of the inclusive (forward) KL divergence to fit a variational distribution that is parameterized by a neural network. Our convergence result relies on the neural tangent kernel (NTK) to characterize the gradient dynamics that arise from considering the variational objective in function space. In the asymptotic regime of a fixed, positive-definite neural tangent kernel, we establish conditions under which the variational objective admits a unique solution in a reproducing kernel Hilbert space (RKHS). Then, we show that the gradient descent dynamics in function space converge to this unique function. We empirically demonstrate that our theoretical results explain the good performance of NPE in non-asymptotic finite-neuron settings.

## 1. Introduction

In variational inference (VI), the parameters $\eta$ of an approximation to the posterior $Q(\Theta; \eta)$ are selected to optimize an objective function, typically the evidence lower bound (ELBO) (Blei et al., 2017). However, the ELBO is generally nonconvex in $\eta$, even for simple variational families such as the family of Gaussian distributions, and so only convergence to a local optimum of the ELBO can be guaranteed (Ghadimi and Lan, 2015; Ranganath et al., 2014; Hoffman et al., 2013). As the number of such optima and the degree of suboptimality of each is generally unknown, the lack of global convergence guarantees constitutes a significant complication for practitioners and a longstanding barrier to the broader adoption of VI. In this work, we present the first global convergence result for variational inference. We accomplish this in the context of an increasingly popular alternative objective for variational inference, the expected forward KL divergence:

$$L_P(\phi) := \mathbb{E}_{P(X)}\text{KL}\left[P(\Theta \mid X) \,||\, Q(\Theta; f(X; \phi))\right]. \tag{1}$$

Here, $P(X)$ denotes a marginal of the model and $P(\Theta \mid X)$ denotes the posterior. For each $x \in \mathcal{X}$, the approximation $Q(\Theta; \eta)$ to $P(\Theta \mid X = x)$ is indexed by distributional parameters $\eta \in \mathcal{Y} \subseteq \mathbb{R}^q$, which are themselves the output of a neural network $f(x; \phi)$ with weights $\phi \in \Phi$. Minimization of this objective is straightforward: computing unbiased gradients requires only sampling $\theta, x \sim P(\Theta, X)$ from the joint model (Section 2.1), readily accomplished by ancestral sampling of $P(\Theta)$ followed by $P(X \mid \Theta)$. Analysis of the amortized problem (i.e.,

optimizing an objective that averages over $P(X)$) is beneficial when considering the forward KL; for the non-amortized problem in which a single observation $x$ is considered, only biased estimates of the gradient of the forward KL can be obtained using self-normalized importance sampling, making convergence difficult to establish (Bornschein and Bengio, 2015; Le et al., 2019; Owen, 2013).

Our analysis considers a functional form of the variational objective $L_P$, given by

$$L_F(f) := \mathbb{E}_{P(X)}\mathrm{KL}\left[P(\Theta \mid X) \,\|\, Q(\Theta; f(X))\right], \tag{2}$$

where $L_F : \mathcal{H} \to \mathbb{R}$ is defined over a general reproducing kernel Hilbert space of functions $\mathcal{H}$. We refer to (1) as the "parametric objective", as its argument is the parameters $\phi \in \Phi$, and we refer to (2) as the "functional objective" as its argument is a function $f \in \mathcal{H}$. These objectives are closely related: under a given network parameterization, provided $f(\cdot; \phi) \in \mathcal{H}$, we have $L_P(\phi) = L_F(f(\cdot; \phi))$. We first demonstrate strict convexity of the functional objective $L_F$ when $Q$ is parameterized as an exponential family distribution (Section 3). This implies the existence of a unique global optimizer $f^*$ of $L_F$ for a large class of variational families. Afterward, we analyze kernel gradient flow dynamics using the neural tangent kernel to show that minimization of $L_P$ results (asymptotically) in an empirical mapping $f$ that is at most $\epsilon$-suboptimal relative to $f^*$ provided a sufficiently flexible neural network is used to parameterize $f$ (Section 4). Together, these results imply that in the infinite-width limit, optimization of $L_P$ by gradient descent recovers a unique global solution.

Our analysis relies on fairly mild conditions, the most important of which are the positive-definiteness of the neural tangent kernel and the structure of the variational family (i.e., an exponential family) (Section 6). Our proofs further assume a two-layer ReLU network architecture for simplicity, but we conjecture that this assumption can be relaxed. Even for practitioners interested in non-amortized inference for a single observable $x_0$, our approach still yields a unique solution. Lifting the problem into the amortized setting yields a unique mapping $f$, and a unique variational approximation for the $x_0$ of interest: $Q(\Theta; f(x_0; \phi))$. Our results thus have implications for practitioners of both amortized and non-amortized VI. Beyond suggesting the advantages of the expected forward KL objective in particular, our results suggest, surprisingly, that a likelihood-free approach to inference can outperform likelihood-based optimization of the ELBO.

## 2. Background

### 2.1. The Expected Forward KL Divergence

The expected forward KL objective is equivalent to the "sleep" objective of Reweighted Wake-Sleep (RWS) (Bornschein and Bengio, 2015), and to the objective considered in forward amortized variational inference (FAVI) (Ambrogioni et al., 2019). These methods in turn fit into the framework of the thermodynamic variational objective (TVO) as a special case. Similar objectives have been considered by neural posterior estimation (NPE) methods (e.g. Papamakarios and Murray (2016); Papamakarios et al. (2019)), but in these methods the prior used to simulate observations (and thus the marginal $P(X)$) mutates during training. Objectives based on the forward KL divergence generally result in varia-

tional posteriors that are overdispersed, a desirable property compared to reverse KL-based optimization (Le et al., 2019; Domke and Sheldon, 2018).

Other VI methods optimize different expectations of the forward KL than Equation (1), typically $\mathbb{E}_{X \sim \mathcal{D}} \mathrm{KL} \left[ P(\Theta \mid X) \mid\mid Q(\Theta; f(X; \phi)) \right]$ (Zhang et al., 2023; McNamara et al., 2024). Here, the outer expectation is over an empirical dataset $\mathcal{D}$ rather than $P(X)$. Averaging over simulated draws from $P(X)$ compared to averaging over the dataset $\mathcal{D}$ is advantageous because when $P(X)$ is used, the resulting method is likelihood-free and admits unbiased gradient estimates (see Appendix B). In non-amortized cases or when the expected forward KL is computed over the dataset $\mathcal{D}$, approximations are required that result in biased gradient estimates (e.g., by self-normalized importance sampling in the wake phase of RWS), for which stochastic gradient descent carries no convergence guarantees.

## 2.2. The Neural Tangent Kernel

A neural network architecture and the parameter space $\Phi$ of its weights together define a family of functions $\{f(\cdot; \phi) : \phi \in \Phi\}$. Let $\ell(x, f(x))$ denote a general real-valued loss function and consider selecting the parameters $\phi$ to minimize $\mathbb{E}_{P(X)} \ell(X, f(X; \phi))$, where $P(X)$ is a fixed distribution on the data space $\mathcal{X}$. The neural tangent kernel (NTK) (Jacot et al., 2018) analyzes the evolution of the function $f(\cdot; \phi)$ while $\phi$ is fitted to minimize the objective above by gradient descent. Continuous-time dynamics are used and $\phi(t)$ and $f(\cdot; \phi(t))$ are defined across continuous time $t$. The parameters $\phi$ thus follow the ODE

$$\dot{\phi}(t) = -\nabla_\phi \mathbb{E}_{P(X)} \ell(X, f(X; \phi(t))). \tag{3}$$

Here, $\dot{\phi}$ denotes the derivative with respect to $t$, and by the chain rule the function values $f(x; \phi(t))$ evolve via

$$\dot{f}(x; \phi(t)) = -\mathbb{E}_{P(X)} \underbrace{J_\phi f(x; \phi(t)) J_\phi f(X; \phi(t))^\top}_{\text{NTK}} \ell'(X, f(X; \phi(t))).$$

Above, we set $\ell'(X, f(X)) := \nabla_f \ell(X, f(X))$ to simplify notation. The product of Jacobians above is known as the *neural tangent kernel* (NTK). The seminal work of Jacot et al. (2018) defined and studied this kernel, given by

$$K_\phi(x, x') = J_\phi f(x; \phi) J_\phi f(x'; \phi)^\top, \tag{4}$$

and established convergence results for certain neural network architectures as the width grows large.

## 3. Convexity of the Functional Objective

We now turn to analysis of the functional objective $L_F$ given in Equation (2). We fix an RKHS $\mathcal{H}$ over which to minimize $L_F$ for now, specializing to the particular choice of $\mathcal{H}$ based on the neural tangent kernel subsequently. Let $\ell(x, f(x)) = \mathrm{KL} \left[ P(\Theta \mid X = x) \mid\mid Q(\Theta; f(x)) \right]$. The functional $L_F$ then has the form $L_F(f) = \mathbb{E}_{P(X)} \ell(X, f(X))$; we will use this form subsequently for neural tangent kernel analysis. Our first result shows that targeting $L_F$ is highly desirable theoretically: $L_F$ admits a unique global minimizer if the variational family $Q$ is an exponential family, as is common practice in VI.

**Lemma 1** *Suppose that $Q(\Theta; \eta)$ is an exponential family distribution with natural parameters $\eta$, sufficient statistics $T(\theta)$, and density $q(\theta; \eta)$ with respect to Lebesgue measure $\lambda(\Theta)$. Then, for any observation $x \in \mathcal{X} \subseteq \mathbb{R}^d$, the loss function*

$$\ell(x, \eta) = \text{KL}\left[P(\Theta \mid X = x) \mid\mid Q(\Theta; \eta)\right]$$

*is strictly convex in $\eta$, provided that $P(\Theta \mid X = x) \ll Q(\Theta; \eta) \ll \lambda(\Theta)$ for all $\eta \in \mathcal{Y} \subseteq \mathbb{R}^q$.*

A proof of Lemma 1 is provided in Appendix A. Lemma 1 shows strict convexity of the function $\ell$ in $\eta$. This immediately implies strict convexity of the functional $L_F(f) = \mathbb{E}_{P(X)}\ell(X, f(X))$ in $f$ by linearity of expectation, which in turn implies the existence of at most one global minimizer.

**Corollary 2** *Suppose that $Q(\Theta; \eta)$ is an exponential family distribution. Then, under the conditions of Lemma 1, the functional objective*

$$L_F(f) := \mathbb{E}_{P(X)}\text{KL}\left[P(\Theta \mid X) \mid\mid Q(\Theta; f(X))\right]$$

*is strictly convex in $f$. Consequently, the set of global minimizers of $L_F$ is either a singleton set or empty.*

We will assume the existence of $f^*$ so that minimization of $L_F$ is well-posed, and also assume $||f^*||_{\mathcal{H}} < \infty$ so that $f^* \in \mathcal{H}$. Hereafter, we shall use "unique" to mean unique almost everywhere with respect to $P(X)$. Furthermore, in a slight abuse of notation, $f^*$ will denote the unique equivalence class of functions that minimizes $L_F$.

Lemma 1 establishes convexity of the (non-amortized) forward KL divergence. Corollary 2 establishes the convexity of $L_F$, the amortized objective, in function space. Convexity holds regardless of the distribution chosen for the outer expectation (e.g., a mixture of point masses corresponding to a empirical dataset may be used, such as in the methods described in Section 2.1).

## 4. Global Optima of the Parametric Objective

In the second phase of our analysis, we consider converging to $f^*$ by gradient descent. As done in practice, we target $L_P$, as optimizing $L_F$ directly is not tractable. We consider performing gradient descent on $\phi$ in continuous time as in Equation (3). Continuous-time dynamics simplify theoretical analysis; stochastic gradient descent with unbiased gradients follows a (noisy) Euler discretization of the continuous ODE (Santambrogio, 2017; Yang et al., 2021). Considering $X \sim P(X)$ for the outer expectation in both $L_P$ and $L_F$ is key in this context: this choice enables unbiased stochastic gradient estimation for $L_P$ (see Appendix B), whereas other choices require approximations that result in biased gradient estimates (see Section 2.1) and thus follow different gradient dynamics.

We focus on a scaled two-layer ReLU network for our results (this architecture is detailed in Appendix C) and use this simple architecture to prove results as the network width $p$ tends to infinity. Our results may be extended to multilayer perceptrons with other activation functions as well. Recall the NTK $K_\phi^p$ from Equation (4), where we now let $p$ denote the network width. We allow multidimensional natural parameters $\eta \in \mathbb{R}^q$ in

our formulation and so for any $p, \phi, x, \tilde{x}$ we have $K_\phi^p(x, \tilde{x}) \in \mathbb{R}^{q \times q}$ because if dim $\Phi = r$, then $J_\phi f(x; \phi) \in \mathbb{R}^{q \times r}$ and so $K_\phi^p(x, x') \in \mathbb{R}^{q \times q}$. For certain neural network architectures, Jacot et al. (2018) show that as the network width $p$ tends to infinity, the neural tangent kernel becomes stable and tends (pointwise) towards a fixed, positive-definite limiting neural tangent kernel $K_\infty$. We prove this convergence holds *uniformly* over the data space $\mathcal{X}$ in Appendix D for our two-layer ReLU architecture. Hereafter, $\mathcal{H}$ is taken to be the RKHS with kernel $K_\infty$.

We bridge the divide between the minimizers of the convex functional $L_F$ and the nonconvex $L_P$ using the limiting kernel. Section 2.2 shows that optimizing $L_P$ causes the network function to evolve according to a *kernel gradient flow* via the neural tangent kernel, i.e. for any fixed $x$, optimization of $L_P$ yields

$$\dot{f}(x; \phi(t)) = -\mathbb{E}_{P(X)} K_{\phi(t)}^p(x, X) \ell'(X, f(X; \phi(t))),$$

where $\dot{f}$ denotes the time derivative. Recalling that $L_F$ has a unique minimizer $f^*$ (Corollary 2), we show that under mild conditions on the limiting tangent kernel $K_\infty$, $f^*$ is the solution obtained by following the same kernel gradient flow dynamics in $\mathcal{H}$ with respect to the limiting neural tangent kernel, i.e. the ODE given by

$$\dot{f}_t(x) = -\mathbb{E}_{P(X)} K_\infty(x, X) \ell'(X, f_t(X)).$$

In other words, beginning from some function $f_0$, following the *limiting* NTK gradient flow dynamics above minimizes the loss functional $L_F$ for sufficiently large $T$. Appendix E provides a proof of Lemma 3 and enumerates regularity conditions (E1)–(E3).

**Lemma 3** *Let $f^*$ denote the minimizer of $L_F$ from Lemma 1, and $\epsilon > 0$. Fix $f_0$, and let $K_\infty$ denote the limiting neural tangent kernel. Let $f_0$ evolve according to the dynamics*

$$\dot{f}_t(x) = -\mathbb{E}_{P(X)} K_\infty(x, X) \ell'(X, f_t(X)).$$

*Suppose that the conditions of Lemma 1 and (E1)-(E3) hold. Then, there exists $T > 0$ such that $L_F(f_T) \le L_F(f^*) + \epsilon$.*

This result enables comparison of the minimizers of $L_P$ and $L_F$ by comparing the two gradient flows above, i.e. kernel gradient flow dynamics that follow $K_{\phi(t)}^p$ and $K_\infty$, respectively. We show that for any fixed $T$, the functions obtained by following kernel gradient dynamics with $K_{\phi(t)}^p$ and $K_\infty$ can be made arbitrarily close to one another, provided $p$ is sufficiently large. This suggests that for large $p$, the gradient descent solution to $L_P$ becomes close to the unique solution $f^*$ of $L_F$. We prove that this is the case in Theorem 4, proven in Appendix E. Regularity conditions (C1)–(C4), (D1)–(D4), and (E1)–(E5) are provided in Appendices C, D, and E, respectively.

**Theorem 4** *Consider the width-$p$ scaled 2-layer ReLU network, evolving via the flow*

$$\dot{f}_t(x) = -\mathbb{E}_{P(X)} K_{\phi(t)}^p(x, X) \ell'(X, f_t(X)), \tag{5}$$

where $f_t$ denotes $f(\cdot, \phi(t))$. Let $f^*$ denote the unique minimizer of $L_F$ from Lemma 1, and fix $\epsilon > 0$. Then, under conditions (C1)–(C4),(D1)–(D4), and (E1)–(E5), there exists $T > 0$ such that almost surely

$$\left[\lim_{p \to \infty} L(f_T)\right] \le L(f^*) + \epsilon, \tag{6}$$

where $L$ is the loss functional $L_F$.

The proof first selects a $T$ by Lemma 3, and then bounds the difference in the trajectories on $[0, T]$ for sufficiently large width $p$ by convergence of the kernels $K^p_{\phi(t)} \to K_\infty$. The proof differs from previous results in that it relies on *uniform* convergence of kernels (cf. Appendices C and D), enabling analysis of population quantities such as $\mathbb{E}_{P(X)}\ell(X, f(X))$. Theorem 4 implies convergence to a unique solution when optimizing $L_P$, despite the highly nonconvex nature of this optimization problem in the network parameters $\phi$. For sufficiently flexible network architectures, optimization of $L_P$ behaves similarly to that of $L_F$, which we have shown is a convex problem in function space $\mathcal{H}$.

## 5. Simulation Study

Here we assess whether the asymptotic regime of Theorem 4 is relevant to practice (with finite width $p$). We explore a diagnostic from Chizat et al. (2019), who provide the intuition that in the limiting NTK regime, the function $f$ behaves much like its linearization around the initial weights $\phi_0$, i.e.,

$$f(x; \phi) \approx f(x; \phi_0) + J_\phi f(x; \phi_0)(\phi - \phi_0). \tag{7}$$

Liu et al. (2020) prove that equality holds exactly in the equation above if and only if $f(x; \phi)$ has a constant tangent kernel (i.e. $K_\infty$). Note that even if $f$ is linear in $\phi$, as in the above expression, it may still be highly nonlinear in $x$. We consider a toy example for which $\|x\|_2 = 1$, a condition assumed for many NTK-based results. The generative model first draws a rotation angle $\Theta$ uniformly between 0 and $2\pi$, and then a rotation perturbation $Z \sim \mathcal{N}(0, \sigma^2)$, where we take $\sigma = 0.5$. Conditional on $\Theta$ and $Z$, the data $x$ is deterministic: $x = [\cos(\theta + z), \sin(\theta + z)]^\top$. This construction ensures that the data lie on the sphere $\mathbb{S}^1 \subset \mathbb{R}^2$, guaranteeing positivity of the limiting NTK for certain architectures (Jacot et al., 2018). We aim to infer $\Theta$ given a realization $x$, marginalizing out the nuisance latent variable $Z$. Our variational family $Q(\Theta; f(x))$ is a von Mises distribution on the interval $[0, 2\pi]$. This family is an exponential family distribution to allow application of Lemma 1. The encoder network $f(\cdot, \phi)$ is given by a two-layer (equivalently, single hidden layer) dense network with rectified linear unit (ReLU) activation, which we study as the network width $p$ grows. The network outputs $f(x; \phi)$ parameterize the natural parameter $\eta$.

We demonstrate that finite $p$ is well described by the asymptotic regime by fitting the neural network $f(x; \phi)$ above, and comparing the results to fitting its linearization $f_{\text{lin}}(x; \phi) = f(x; \phi_0) + J_\phi f(x; \phi_0)(\phi - \phi_0)$ in $\phi$ for differing widths $p$. For both settings, stochastic gradient estimation was performed by following the procedure in Appendix B. For evaluation, we fix $N = 1000$ independent realizations $x_1^*, \ldots, x_N^*$ from the generative model with underlying ground-truth latent parameter values $\theta_1^*, \ldots, \theta_N^*$, and evaluate the

held-out negative log-likelihood (NLL), $-\frac{1}{N}\sum_{i=1}^{N}\log q\left(\theta_i^*; f(x_i^*; \phi)\right)$, for each of the two functions $f(x; \phi)$ and $f_{\mathrm{lin}}(x; \phi)$. Figure 1 shows the evolution of the held-out NLL across the fitting procedure for three different network widths $p$: 64, 256, and 1024. The difference in quality between the linearized and true functions at convergence diminishes as the width $n$ grows; for $n = 1024$ the two are nearly identical, providing evidence that the asymptotic regime of Section 4 is achieved, i.e. that the neural tangent kernel is approximately $K_\infty$.

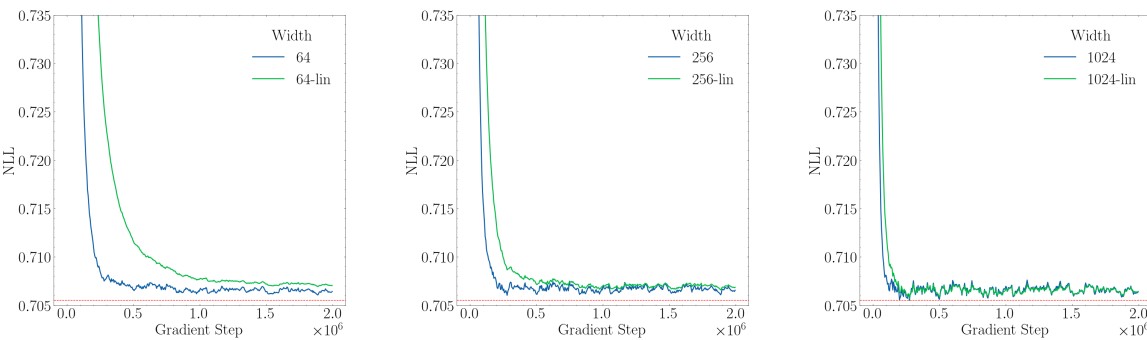

Figure 1: Negative log-likelihood across gradient steps, for network widths 64, 256, and 1024 neurons. NLL for the exact posterior is denoted by the red line.

## 6. Discussion

In this work, we showed that in the asymptotic limit of an infinitely wide neural network, gradient descent dynamics on the expected forward KL objective $L_P$ converge to an $\epsilon$ neighborhood of a unique function $f^*$. The proofs of Lemma 1 and Theorem 4 depend on several regularity conditions, the most important of which is the positive definiteness of the limiting neural tangent kernel (Section 2.2) and the compactness of the data space $\mathcal{X}$. The former allows us to take $\mathcal{H}$, the domain of $L_F$, to be the RKHS defined by the limiting neural tangent kernel, while the latter is key in establishing *uniform* convergence of kernels on $\mathcal{X}$, necessary to analyze population losses such as $L_F$ and $L_P$ that take the form of expectations with respect to a continuous measure on $\mathcal{X}$.

Like other neural tangent kernel results, we analyze a specific architecture and initialization of the network weights, and require smoothness of the objective. Our results consider a single hidden-layer ReLU network, as is common in the NTK literature, and are asymptotic in its width $p$. We show experimentally that the asymptotic regime accurately characterizes training dynamics in practice for a finite number of neurons (Section 5), which provides evidence that our global convergence result describes training dynamics in empirical settings, i.e. the convex behavior of $L_P$-based minimization is achieved in practice.

Expected forward KL minimization is a likelihood-free inference (LFI) method. This class of methods is often viewed as a tool only for use in the case where the likelihood function is unavailable. Our results suggest, instead, that likelihood-free approaches to inference may be preferable even when the likelihood function is readily available. Likelihood-based optimization of the ELBO often converges to a shallow local optimum, while, under reasonable conditions, expected forward KL minimization converges to a global optimum.

## Acknowledgments

This material is based upon work supported by the National Science Foundation under Grant Nos. 2209720 (OAC) and 1841052 (DGE). We thank the reviewers for their helpful comments and suggestions about this manuscript.

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

## Appendix A.  Convexity of the Functional Objective

We prove Lemma 1 from the manuscript below.

**Lemma 1** *Suppose that $Q(\Theta; \eta)$ is an exponential family distribution with natural parameters $\eta$, sufficient statistics $T(\theta)$, and density $q(\theta; \eta)$ with respect to Lebesgue measure $\lambda(\Theta)$. Then, for any observation $x \in \mathcal{X} \subseteq \mathbb{R}^d$, the loss function*

$$\ell(x, \eta) = \mathrm{KL}\left[P(\Theta \mid X = x) \mid\mid Q(\Theta; \eta)\right]$$

*is strictly convex in $\eta$, provided that $P(\Theta \mid X = x) \ll Q(\Theta; \eta) \ll \lambda(\Theta)$ for all $\eta \in \mathcal{Y} \subseteq \mathbb{R}^q$.*

**Proof** Let the log-density be given by $\log q(\theta; \eta) = \log h(\theta) + \eta^\top T(\theta) - A(\eta)$. First observe that under the conditions given, the function $\ell$ is equivalent (up to additive constants) to a much simpler expression, the expected log-density of $q$, via

$$\begin{aligned}
\ell(x, \eta) &= \mathrm{KL}\left[P(\Theta|X = x) \mid\mid Q(\Theta; \eta)\right] \\
&= H_x - \mathbb{E}_{P(\Theta|X=x)} \log q(\theta; \eta) \\
&\stackrel{\eta}{=} -\mathbb{E}_{P(\Theta|X=x)} \log q(\theta; \eta).
\end{aligned}$$

Now, the mapping $\eta \to -\log q(\theta; \eta)$ is convex in $\eta$ because its Hessian is $\frac{\partial A}{\partial \eta \partial \eta^\top} = \mathrm{Var}(T(\theta)) \succ 0$ (cf. Chapter 6.6.3 of Srivastava et al. (2014)). We can show $\ell$ is convex in $\eta$ by applying linearity of expectation. We have for any $\lambda \in [0, 1]$

$$\ell(x, \lambda \eta_1 + (1 - \lambda)\eta_2) = -\mathbb{E}_{P(\Theta|X=x)} \log q(\Theta; \lambda \eta_1 + (1 - \lambda)\eta_2) \tag{8}$$

$$\leq -\mathbb{E}_{P(\Theta|X=x)}\left(\lambda \log q(\Theta; \eta_1) + (1 - \lambda) \log q(\Theta; \eta_2)\right) \tag{9}$$

$$= \lambda \ell(x, \eta_1) + (1 - \lambda)\ell(x, \eta_2) \tag{10}$$

where the second line follows from convexity of the map $\eta \mapsto -\log q(\theta; \eta)$ above for any value of $\theta$. So the function $\ell(x, \eta)$ is strictly convex in $\eta$. ∎

## Appendix B.  Unbiased Stochastic Gradients for the Parametric Objective

Computation of unbiased estimates of gradient of the loss function $L_P(\phi)$ with respect to parameters $\phi$ is all that is needed to implement SGD for $L_P$. Under mild conditions (see Proposition 5), the loss function $L_P(\phi)$ may be equivalently written as

$$L_P(\phi) = \mathbb{E}_{P(X)}\mathbb{E}_{P(\Theta|X)} \log \frac{p(\Theta \mid X)}{q(\Theta; f(X; \phi))} = \mathbb{E}_{P(\Theta, X)} \log \frac{p(\Theta \mid X)}{q(\Theta; f(X; \phi))}$$

for density functions $p, q$, where $f(\cdot, \phi)$ denotes a function parameterized by $\phi$. Under the conditions of Theorem 5 differentiation and integration may be interchanged, so that

$$\nabla_\phi L_P(\phi) = \mathbb{E}_{P(\Theta, X)} \nabla_\phi \log \frac{p(\Theta \mid X)}{q(\Theta; f(X; \phi))} = -\mathbb{E}_{P(\Theta, X)} \nabla_\phi \log q(\Theta; f(X; \phi))$$

and unbiased estimates of the quantity can be easily attained by samples drawn $(\theta, x) \sim P(\Theta, X)$.

**Proposition 5** *Let $(\Omega_1, \mathcal{B}_1)$, $(\Omega_2, \mathcal{B}_2)$ be measurable spaces on which the random variables $X : \Omega_1 \to \mathcal{X}$ and $\Theta : \Omega_2 \to \mathcal{O}$ are defined, respectively. Suppose that for all $x \in \mathcal{X}$ and all $\phi \in \Phi$ we have $P(\Theta \mid X = x) \ll Q(\Theta; f(x; \phi)) \ll \lambda(\Theta)$, with $\lambda(\Theta)$ denoting Lebesgue measure and $\ll$ denoting absolute continuity. Further, suppose that $\log\left(\frac{p(\Theta|X)}{q(\Theta;f(X;\phi))}\right)$ is measurable with respect to the product space $(\Omega_1 \times \Omega_2, \mathcal{B}_1 \times \mathcal{B}_2)$ for each $\phi \in \Phi$, and $\nabla_\phi \log q(\theta; f(x; \phi))$ exists for almost all $(\theta, x) \in \mathcal{O} \times \mathcal{X}$. Finally, assume there exists an integrable $Y$ dominating $\nabla_\phi \log q(\theta; f(x; \phi))$ for all $\phi \in \Phi$ and almost all $(\theta, x) \in \mathcal{O} \times \mathcal{X}$. Then for any $B \in \mathbb{N}$ and any $\phi \in \Phi$ the quantity*

$$\hat{\nabla}(\phi) = -\frac{1}{B} \sum_{i=1}^{B} \nabla_\phi \log q(\theta_i; f(x_i; \phi)), \quad (\theta_i, x_i) \overset{iid}{\sim} P(\Theta, X) \qquad (11)$$

*is an unbiased estimator of the gradient of the objective $L_P$, evaluated at $\phi \in \Phi$.*

**Proof** By the absolute continuity assumptions, for any $x \in \mathcal{X}$ the distributions $P(\Theta \mid X = x)$ and $Q(\Theta; f(x; \phi))$ admit densities with respect to Lebesgue measure denoted $p(\theta \mid x)$ and $q(\theta; f(x; \phi))$, respectively. We may then rewrite the KL divergence from Equation (1) as

$$\mathrm{KL}\left[ P(\Theta \mid X = x) \| Q(\Theta; f(x; \phi)) \right] := \mathbb{E}_{P(\Theta|X=x)} \log\left( \frac{dP(\Theta \mid X = x)}{dQ(\Theta; f(x; \phi))} \right)$$

$$= \mathbb{E}_{P(\Theta|X=x)} \log\left( \frac{p(\Theta \mid x)}{q(\Theta; f(x; \phi))} \right)$$

because the Radon-Nikodym derivative $dP/dQ$ is given by the ratio of these densities. Equation (1) is thus equivalent to

$$\mathbb{E}_{P(X)} \mathbb{E}_{P(\Theta|X)} \log\left( \frac{p(\Theta \mid X)}{q(\Theta; f(X; \phi))} \right) = \mathbb{E}_{P(\Theta,X)} \log\left( \frac{p(\Theta \mid X)}{q(\Theta; f(X; \phi))} \right).$$

This expectation is well-defined by the measurability assumption on $\log\left(\frac{p(\Theta|X)}{q(\Theta;f(X;\phi))}\right)$. To interchange differentiation and integration, it suffices by Leibniz's rule that the gradient of this quantity with respect to $\phi$ is dominated by a measurable r.v. $Y$. More precisely, there exists integrable $Y(\theta, x)$ defined on the product space $\mathcal{O} \times \mathcal{X}$ such that $\left|\left| \nabla_\phi \log\left(\frac{p(\theta|x)}{q(\theta;f(x;\phi))}\right) \right|\right| \leq Y(\theta, x)$ for all $\phi \in \Phi$ and almost everywhere-$P(\Theta, X)$. This is assumed in the statement of the proposition, and so we have

$$\nabla_\phi \mathbb{E}_{P(\Theta,X)} \log\left( \frac{p(\Theta \mid X)}{q(\Theta; f(X; \phi))} \right) = \mathbb{E}_{P(\Theta,X)} \nabla_\phi \log\left( \frac{p(\Theta \mid X)}{q(\Theta; f(X; \phi))} \right)$$

$$= -\mathbb{E}_{P(\Theta,X)} \nabla_\phi \log q(\Theta; f(X; \phi))$$

and the result follows by sampling.  ∎

The variance of the gradient estimator can be reduced at the standard Monte Carlo rate, and for any $B$ Equation (11) can be used for stochastic gradient descent (SGD).

## Appendix C. The Limiting NTK

Before proceeding, we introduce the architecture specific to our analysis, a scaled two-layer network, and several theorems that we will use throughout the analysis.

The first result from Shapiro (2003) concerns optimization of the objective $f(x) = \mathbb{E}_{\xi \sim P} F(x; \xi)$ in $x$ via its empirical approximation $\hat{f}_n(x) = \frac{1}{n} \sum_{i=1}^{n} F(x; \xi_i), \xi_i \overset{iid}{\sim} P$. We reproduce this result below.

**Theorem 6 (Proposition 7 of Shapiro (2003))** *Let $C$ be a nonempty compact subset of $\mathbb{R}^n$ and suppose that (i) for almost every $\xi \in \Xi$ the function $F(\cdot, \xi)$ is continuous on $C$, (ii) $F(x, \xi)$, $x \in C$, is dominated by an integrable function, (iii) the sample $\xi_1, \ldots, \xi_n$ is iid. Then the expected value function $f(x)$ is finite valued and continuous on $C$, and $\hat{f}_n(x)$ converges to $f(x)$ with probability 1 uniformly on $C$.*

The next two results are integral forms of Gronwall's inequality that we use in subsequent analysis. We refer to Dragomir (2003) for a detailed review, and present simplified versions of the results therein below.

**Theorem 7 (Gronwall's Inequality, Corollary 3 of Dragomir (2003))** *Let $u(t) \in \mathbb{R}$ be such that $u(t) \le c_1 + c_2 \int_0^t u(s) ds$ for $t > 0$ and nonnegative $c_1, c_2$. Then*

$$u(t) \le c_1 \exp[c_2 t].$$

**Theorem 8 (Theorem 57 of Dragomir (2003))** *Let $u(t) \in \mathbb{R}$ be such that $u(t) \le c_1 + c_2 \int_0^t \int_0^s u(v) dv ds$ for $t > 0$ and nonnegative $c_1, c_2$. Then*

$$u(t) \le c_1 \exp[c_2 t^2 / 2].$$

Now we turn to specifics of the architecture we consider. Assume the function $f$ has the architecture of a (scaled) two-layer (single hidden layer) network mapping $f : \mathcal{X} \to \mathcal{Y}$ with $\mathcal{X} \subseteq \mathbb{R}^d$ and $\mathcal{Y} \subseteq \mathbb{R}^q$. We consider this network architecture for a given width $p$, and study each of the $i = 1, \ldots, q$ coordinate functions of $f$. For a scaled two-layer network, the $i$th such function is

$$f_i(x; \phi) := \frac{1}{\sqrt{p}} \sum_{j=1}^{p} a_{ij} \sigma \left( x^\top w_j \right)$$

for $i = 1, \ldots, q$, where $\sigma$ denotes an activation function. The scaling depends on the width of the network $p$. The parameters $\phi$ are thus $\phi = \{w_j, a_{(\cdot), j}\}_{j=1}^{p}$ where $a_{(\cdot), j}$ denotes the vector $[a_{1j}, \ldots, a_{qj}]^\top$ (i.e. the $j$th coefficient for each component function $i$). The individual parameters have dimensions as follows: $w_j \in \mathbb{R}^d$ and $a_{(\cdot), j} \in \mathbb{R}^q$, for all $j = 1, \ldots, p$, where again $p$ denotes the network width and $d$ the data dimension dim $\mathcal{X}$. For ease hereafter, we write $a_j = a_{(\cdot), j}$ to refer to the entire $j$th vector of second layer network coefficients when the context is clear. As is standard, the first layer parameters are initialized as independent standard Gaussian random variables, i.e. $w_j \overset{iid}{\sim} \mathcal{N}(0, I_d)$ for all $j = 1, \ldots, p$. The weight $a_{ij}$ can also be drawn $a_{ij} \overset{iid}{\sim} \mathcal{N}(0, 1)$ for all $i = 1, \ldots, q, j = 1, \ldots, p$, but in this work we initialize these second-layer weights to zero for simplicity to ensure that at initialization, $f(\cdot; \phi) = 0$. A zero-initialized network function is used for analysis in several related works,

e.g. Chizat et al. (2019) and Ba et al. (2020). For now, notationally we denote weights to be initialized as draws from an arbitrary distribution $D$, and we introduce specificity to $D$ as required.

The neural tangent kernel (Equation (4)) can be computed explicitly for this architecture with ease, and is given in the lemma below which proves pointwise convergence to the limiting NTK at initialization as the width $p$ tends to infinity.

**Lemma 9 (Pointwise Convergence At Initialization)** *For the architecture above, consider any $p$. Let $a \in \mathbb{R}^q, w \in \mathbb{R}^d$ be distributed according to $a, w \sim D$ for some distribution $D$ such that $a, w$ are integrable ($L_1$) random variables. Assume $\mathcal{X}$ is compact, and $\sigma'$ is bounded. Then provided condition (C4) holds (see below), we have for any $x, \tilde{x} \in \mathcal{X}$ that*

$$K^p_{\phi(0)}(x, \tilde{x}) \overset{a.s.}{\to} \mathbb{E}_D K(x, \tilde{x}; w, a) \tag{12}$$

*as $p \to \infty$ where $K^p_{\phi(0)}$ denotes the NTK at initialization constructed from draws $a_j, w_j \overset{iid}{\sim} D$ and $K(x, \tilde{x}; w, a) \in \mathbb{R}^{q \times q}$ is the $q \times q$ matrix whose $k, l$th entry is given by*

$$\left[ \mathbf{1}_{k=l} \sigma\left(x^\top w\right) \sigma\left(\tilde{x}^\top w\right) + a_k a_l \sigma'\left(x^\top w\right) \sigma'\left(\tilde{x}^\top w\right) \left(x^\top \tilde{x}\right) \right]$$

*for $k, l = 1, \ldots, q$.*

**Proof** Consider the $k$th coordinate function of $f$. For any choice of $p$, the gradient is given by

$$\nabla_\phi f_k(x; \phi) = \begin{bmatrix} \frac{\partial f_k(x;\phi)}{\partial a_{k1}} \\ \vdots \\ \frac{\partial f_k(x;\phi)}{\partial a_{kp}} \\ \frac{\partial f_k(x;\phi)}{\partial w_1} \\ \vdots \\ \frac{\partial f_k(x;\phi)}{\partial w_p} \end{bmatrix} = \frac{1}{\sqrt{p}} \begin{bmatrix} \sigma\left(x^\top w_1\right) \\ \vdots \\ \sigma\left(x^\top w_p\right) \\ a_{k1}\sigma'\left(x^\top w_1\right) x \\ \vdots \\ a_{kp}\sigma'\left(x^\top w_p\right) x \end{bmatrix}$$

where we have imposed an arbitrary ordering on the parameters. In the above, we omitted partial derivatives $\frac{\partial f_k}{\partial a_{lj}}$ for $l \neq k, j = 1, \ldots, p$ because these are all identically zero. From this, it follows that for any fixed $x, \tilde{x} \in \mathcal{X}$ we have the $k, l$-th entry of $K^p_{\phi(0)}(x, \tilde{x})$ is given by

$$\nabla_\phi f_k(x; \phi(0))^\top \nabla_\phi f_l(\tilde{x}; \phi(0)) = \frac{1}{p} \sum_{j=1}^p \mathbf{1}_{k=l} \sigma\left(x^\top w_j\right) \sigma\left(\tilde{x}^\top w_j\right) +$$

$$\frac{1}{p} \sum_{j=1}^p a_{kj} a_{lj} \sigma'\left(x^\top w_j\right) \sigma'\left(\tilde{x}^\top w_j\right) \left(x^\top \tilde{x}\right).$$

The existence of the limiting NTK follows immediately: for each of the two terms above, each term is clearly integrable by compactness of $\mathcal{X}$ and domination (see (C4)). It follows that $K_\infty(x, \tilde{x})$ (the pointwise limit) is the $q \times q$ matrix whose $k, l$th entry is given by

$$\mathbb{E}_{w,a \sim D} \left[ \mathbf{1}_{k=l} \sigma\left(x^\top w\right) \sigma\left(\tilde{x}^\top w\right) + a_k a_l \sigma'\left(x^\top w\right) \sigma'\left(\tilde{x}^\top w\right) \left(x^\top \tilde{x}\right) \right]$$

with $w, a \sim D$. Convergence in probability pointwise follows from the weak law of large numbers, and almost sure convergence holds by the strong law of large numbers. $K(x, x; a, w)$ is integrable by the assumption (C4) (see below), so the expectation is well-defined. ∎

The proof of the existence and pointwise convergence to the limiting NTK $K_\infty$ above is rather straightforward, and this result has been previously established in other works (Jacot et al., 2018). For our analysis of kernel gradient flows in Theorem 4 for the expected forward KL objectives $L_P$ and $L_F$, however, we require *uniform* convergence to $K_\infty$ over the entire data space $\mathcal{X}$.

We establish conditions under which this uniform convergence holds in two results, Proposition 10 and Proposition 13. Proposition 10, given below, concerns uniform convergence at initialization to the limiting neural tangent kernel $K_\infty$ (i.e. before beginning gradient descent). Proposition 13, proven in Appendix D, demonstrates that across a finite training interval $[0, T]$, the NTK changes minimally from its initial value in a large width regime. Generally, we refer to the first result as "deterministic initialization" and the second as "lazy training" following related works (Jacot et al., 2018; Chizat et al., 2019).

Below, we give suitable regularity conditions and state and prove Proposition 10.

(C1) The data space is $\mathcal{X}$ is compact.

(C2) The distribution $D$ is such that $w \sim \mathcal{N}(0, I_d)$ and $a = 0$ w.p. 1. For $j = 1, \ldots, p$ iid draws from this distribution, we thus have $w_j \overset{iid}{\sim} \mathcal{N}(0, I_d)$ and $a_{ij} = 0$ w.p 1 for all $i, j$.

(C3) The activation function $\sigma$ is continuous. Under (C2), this implies that the function $K(\cdot, \cdot; a, w)$ from Lemma 9 with $a, w \sim D$ is almost surely continuous.

(C4) The function $K(x, \tilde{x}; a, w)$ is dominated by some integrable random variable $G$, i.e. for all $x, \tilde{x} \in \mathcal{X} \times \mathcal{X}$ we have $||K(x, \tilde{x}; a, w)||_F \leq G(a, w)$ almost surely for integrable $G(a, w)$.

**Proposition 10** *Fix a scaled two-layer network architecture of width $p$, and let $\Phi$ denote the corresponding parameter space. Initialize $\phi(0)$ as independent, identically distributed random variables drawn from the distribution $D$ in (C2). Let $K^p_{\phi(0)} : \mathcal{X} \times \mathcal{X} \to \mathbb{R}^{q \times q}$ be the mapping defined by $(x, x') \mapsto K_{\phi(0)}(x, x') = J_\phi f(x; \phi(0)) J_\phi f(x'; \phi(0))^\top$. Then provided conditions (C1)–(C4) hold, we have as $p \to \infty$ that*

$$\sup_{x, \tilde{x} \in \mathcal{X}} ||K^p_{\phi(0)}(x, \tilde{x}) - K_\infty(x, \tilde{x})||_F \overset{a.s.}{\to} 0, \tag{DI}$$

*where $K_\infty(x, \tilde{x}) := \text{plim}_{p \to \infty} K^p_{\phi(0)}(x, \tilde{x})$ is a fixed, continuous kernel.*

**Proof** The proof follows by direct application of Proposition 7 of Shapiro (2003). Precisely, we satisfy i) almost-sure continuity of $K(\cdot, \cdot; a, w)$ by (C3), ii) domination by (C4), and iii) the draws comprising $K^p_{\phi(0)}$ are iid by assumption. By this proposition, then, we have uniform convergence of $K^p_{\phi(0)}$ to $K_\infty$ and get continuity of $K_\infty$ as well. ∎

# Appendix D. Lazy Training

Below, we prove several results that will aid in proving the "lazy training" result of Proposition 13 (see below). Given the same architecture as above in Appendix C and a fixed width $p$ and time $T > 0$, we will begin by bounding $||w_j(T) - w_j(0)||$ and $||a_{kj}(T) - a_{kj}(0)||, ||a_{lj}(T) - a_{lj}(0)||$ for all $k, l = 1, \ldots, q$ and all $j = 1, \ldots, p$. As in Appendix C, there are several conditions that we impose and use in the following results. (D1)–(D2) are identical to (C1)–(C2), repeated for clarity.

(D1) The data space is $\mathcal{X}$ is compact.

(D2) The distribution $D$ is such that $w \sim \mathcal{N}(0, I_d)$ and $a = 0$ w.p. 1. For $j = 1, \ldots, p$ iid draws from this distribution, we thus have $w_j \overset{iid}{\sim} \mathcal{N}(0, I_d)$ and $a_{ij} = 0$ w.p 1 for all $i, j$.

(D3) The function $\ell(x, \eta) = \text{KL}\left[P(\Theta \mid X = x) \,||\, Q(\Theta; \eta)\right]$ is such that $\ell'(x; \eta)$ is bounded uniformly for all $x$ and for all $\eta \in \{f(x; \phi(t)) : t > 0\}$ by a constant $\tilde{M}$, uniformly over the width $p$. We recall that this notation is shorthand for $\nabla_\eta \ell(x, \eta)$.

(D4) The activation function $\sigma(\cdot)$ is non-polynomial and is Lipschitz with constant $C$. Note that the Lipschitz condition implies $\sigma$ has bounded first derivative i.e. $|\sigma'(r)| \leq C$ for all $r \in \mathbb{R}$.

With these conditions in hand, we now prove several lemmas for individual parameters.

**Lemma 11 (Lazy Training of $w$)** *For the width $p$ scaled two-layer architecture above, assume conditions (D1)–(D4) hold. Let $\phi$ evolve according to the gradient flow of the objective $L_P$, i.e.*

$$\dot{\phi}(t) = -\nabla_\phi L_P(\phi).$$

*Fix any $T > 0$. Then for all $j = 1, \ldots, p$ we have almost surely that*

$$||w_j(T) - w_j(0)||_2 \leq ||w_j(0)||_2 \cdot D_{p,T} + E_{p,T} \tag{13}$$

*where $D_{p,T}, E_{p,T}$ are constants depending on $p, T$ and satisfying $\lim_{p \to \infty} D_{p,T} = 0$ and $\lim_{p \to \infty} E_{p,T} = 0$.*

**Proof** First note that for any fixed $j$, we have

$$J_{w_j} f(x; \phi) = \begin{bmatrix} \nabla_{w_j} f_1(x; \phi)^\top \\ \vdots \\ \nabla_{w_j} f_q(x; \phi)^\top \end{bmatrix} = \frac{1}{\sqrt{p}} \begin{bmatrix} a_{1j}\sigma'\left(x^\top w_j\right) x^\top \\ \vdots \\ a_{qj}\sigma'\left(x^\top w_j\right) x^\top \end{bmatrix} \in \mathbb{R}^{q \times d}$$

as required, where $a_{ij} \in \mathbb{R}$ for $i = 1, \dots, q$ and $x \in \mathcal{X} \subseteq \mathbb{R}^d$ from (D1). We can bound the operator 2-norm of this matrix by observing that for any $y \in \mathbb{R}^d$ we have

$$
\begin{aligned}
||J_{w_j} f(x; \phi) y||_2^2 &= \frac{1}{p} \cdot \left( \sum_{i=1}^{q} a_{ij}^2 \right) \cdot \sigma' \left( x^\top w_j \right)^2 (x^\top y)^2 \\
&\leq \frac{C^2}{p} ||a_j||_2^2 \cdot ||x||_2^2 \cdot ||y||_2^2 \quad \text{by (D4) and Cauchy-Schwarz} \\
\implies ||J_{w_j} f(x; \phi)||_2 &\leq \frac{C}{\sqrt{p}} ||a_j||_2
\end{aligned}
$$

by observing $||x||_2^2$ is bounded by (D1) (and we absorb this term into the constant $C$). By similar computations, we also have

$$
J_{a_j} f(x; \phi) = \begin{bmatrix} \nabla_{a_j} f_1(x; \phi)^\top \\ \vdots \\ \nabla_{a_j} f_q(x; \phi)^\top \end{bmatrix} = \frac{1}{\sqrt{p}} \operatorname{diag} \begin{bmatrix} \sigma\left(x^\top w_j\right) \\ \vdots \\ \sigma\left(x^\top w_j\right) \end{bmatrix} \in \mathbb{R}^{q \times q}.
$$

Using condition (D4), it follows that

$$
\begin{aligned}
||J_{a_j} f(x; \phi)||_2 &\leq \frac{|\sigma(x^\top w_j)|}{\sqrt{p}} \\
&\leq \frac{|\sigma(0)| + C|x^\top w_j|}{\sqrt{p}|} \\
&\overset{def}{=\!=} \frac{K + C|x^\top w_j|}{\sqrt{p}} \\
&\leq \frac{K + C||w_j||_2}{\sqrt{p}}
\end{aligned}
$$

by Cauchy-Schwarz and (D1),(D4), where throughout the following we let $K := |\sigma(0)|$ and again absorb the $||x||$ term into the general constant $C$. Now we will use these computations

to bound the variation on $w_j$ across the interval $(0, T]$. Fix any $t \in (0, T]$. Then we have

$$
\begin{aligned}
||w_j(t) - w_j(0)||_2 &\leq \int_0^t ||\dot{w}_j(s)||ds \\
&\leq \int_0^t \mathbb{E}_{P(X)} ||J_{w_j} f(X; \phi(s)) \ell'(X, f(X; \phi(s)))||_2 ds \\
&\leq \tilde{M} \int_0^t \mathbb{E}_{P(X)} ||J_{w_j} f(X; \phi(s))||_2 ds \quad \text{by (D3)} \\
&\leq \frac{C\tilde{M}}{\sqrt{p}} \int_0^t ||a_j(s)||_2 ds \quad \text{by above work} \\
&\overset{a.s.}{=} \frac{C\tilde{M}}{\sqrt{p}} \int_0^t ||a_j(s) - a_j(0)||_2 ds \quad \text{by (D2)} \\
&\leq \frac{C\tilde{M}}{\sqrt{p}} \int_0^t \int_0^s ||\dot{a}_j(v)||_2 dv ds \\
&\leq \frac{C\tilde{M}}{\sqrt{p}} \int_0^t \int_0^s \mathbb{E}_{P(X)} ||J_{a_j} f(X; \phi)||_2 ||\ell'(X, f(X; \phi(v)))||_2 dv ds \\
&\leq \frac{C\tilde{M}^2}{\sqrt{p}} \int_0^t \int_0^s \mathbb{E}_{P(X)} ||J_{a_j} f(X; \phi)||_2 dv ds \quad \text{by (D3)} \\
&\leq \frac{C\tilde{M}^2 K t^2}{2p} + \frac{C^2 \tilde{M}^2}{p} \int_0^t \int_0^s ||w_j(v)||_2 dv ds \quad \text{by above work} \\
&\leq \frac{C\tilde{M}^2 K t^2}{2p} + \frac{C^2 \tilde{M}^2}{p} \int_0^t \int_0^s ||w_j(v) - w_j(0)||_2 + ||w_j(0)||_2 dv ds \\
&= \frac{C\tilde{M}^2 K t^2}{2p} + \frac{C^2 \tilde{M}^2 t^2}{2p} ||w_j(0)||_2 + \frac{C^2 \tilde{M}^2}{p} \int_0^t \int_0^s ||w_j(v) - w_j(0)||_2 dv ds \\
&\leq \frac{C\tilde{M}^2 K T^2}{2p} + \frac{C^2 \tilde{M}^2 T^2}{2p} ||w_j(0)||_2 + \frac{C^2 \tilde{M}^2}{p} \int_0^t \int_0^s ||w_j(v) - w_j(0)||_2 dv ds \\
&= c_1 + \int_0^t \int_0^s c_2 ||w_j(v) - w_j(0)||_2 dv ds
\end{aligned}
$$

with $c_1 = \frac{C\tilde{M}^2 K T^2}{2p} + \frac{C^2 \tilde{M}^2 T^2}{2p} ||w_j(0)||_2$ and $c_2 = \frac{C^2 \tilde{M}^2}{p}$. Note that even though $c_1$ depends on $T$, this is constant as $T$ is fixed. We write these quantities in this way to recognize a Gronwall-type inequality that we can use to bound the left hand side. Indeed, by direct application of Theorem 57 of Dragomir (2003) (see Theorem 8) we have that

$$
\begin{aligned}
||w_j(t) - w_j(0)||_2 &\leq c_1 \exp\left[ \int_0^t \int_0^s c_2 dv ds \right] \\
&= c_1 \exp \frac{c_2 t^2}{2} \\
&= \left( \frac{C\tilde{M}^2 K T^2}{2p} + \frac{C^2 \tilde{M}^2 T^2}{2p} ||w_j(0)||_2 \right) \exp\left[ \frac{C^2 \tilde{M}^2 t^2}{2p} \right].
\end{aligned}
$$

giving the result for $t = T$ if we take $D_{p,T} = \frac{C^2 \tilde{M}^2 T^2}{2p} \exp\left[\frac{C^2 \tilde{M}^2 T^2}{2p}\right]$ and $E_{p,T} = \frac{C\tilde{M}^2 K T^2}{2p} \exp\left[\frac{C^2 \tilde{M}^2 T^2}{2p}\right]$. Clearly these constants satisfy $\lim_{p\to\infty} D_{p,T} = 0$ and $\lim_{p\to\infty} E_{p,T} = 0$ for any fixed $T$.

∎

**Lemma 12 (Lazy Training of $a$)** *Under the same conditions as Lemma 11, let $\phi$ evolve according to the gradient flow of problem $L_P$, i.e.*

$$\dot{\phi}(t) = -\nabla_\phi L_P(\phi).$$

*Fix any $T > 0$. Then we have for any $j$ that*

$$||a_j(T)||_2 \leq ||w_j(0)||_2 \cdot F_{p,T} + G_{p,T} \tag{14}$$

*almost surely, where $E_{p,T}$ and $F_{p,T}$ are constants depending on $p, T$ satisfying $\lim_{p\to\infty} E_{p,T} = 0$ and $\lim_{p\to\infty} F_{p,T} = 0$.*

**Proof** We will use much of the same work as in Lemma 11. Namely, $||a_j(t)||_2 = ||a_j(t) - a_j(0)||_2$ almost surely by (D2), and thereafter for any $t \in (0, T]$ we have

$$
\begin{aligned}
||a_j(t) - a_j(0)||_2 &\leq \int_0^t ||\dot{a}_j(v)||_2 ds \\
&\leq \frac{1}{\sqrt{p}} \int_0^t \mathbb{E}_{P(X)} ||J_{a_j} f(X; \phi)||_2 ||\ell'(X, f(X; \phi(v)))||_2 ds \\
&\leq \frac{\tilde{M}}{\sqrt{p}} \int_0^t \mathbb{E}_{P(X)} ||J_{a_j} f(X; \phi)||_2 ds \\
&\leq \frac{K\tilde{M}t}{p} + \frac{\tilde{M}C}{p} \int_0^t ||w_j(s)||_2 ds \quad \text{by work in Lemma 11} \\
&\leq \frac{K\tilde{M}t}{p} + \frac{\tilde{M}C}{p} \int_0^t ||w_j(s) - w_j(0)||_2 + ||w_j(0)||_2 ds \\
&\leq \frac{K\tilde{M}t}{p} + \frac{\tilde{M}Ct}{p} ||w_j(0)||_2 + \frac{\tilde{M}C}{p} \int_0^t D_{p,s} ||w_j(0)||_2 + E_{p,s} ds \quad \text{a.s. by Lemma 11} \\
&\leq \frac{K\tilde{M}t}{p} + \frac{\tilde{M}Ct}{p} ||w_j(0)||_2 + \frac{\tilde{M}C}{p} \int_0^t E_{p,s} ds + \frac{\tilde{M}C}{p} ||w_j(0)||_2 \int_0^t D_{p,s} ds \\
&= ||w_j(0)||_2 \left(\frac{\tilde{M}Ct}{p} + \frac{\tilde{M}C}{p} \int_0^t D_{p,s} ds\right) + \left(\frac{K\tilde{M}t}{p} + \frac{\tilde{M}C}{p} \int_0^t E_{p,s} ds\right) \\
&\stackrel{def}{=} ||w_j(0)||_2 \cdot F_{p,t} + G_{p,t}
\end{aligned}
$$

Clearly, these constants satisfy $\lim_{p\to\infty} F_{p,t} \to 0$ and $\lim_{p\to\infty} G_{p,t} \to 0$ (to see this, simply plug in the forms of $D_{p,s}$ and $E_{p,s}$ from Lemma 11 above) and we have the result by taking $t = T$.

∎

Now with these results in hand, we may state and prove Proposition 13, given below.

**Proposition 13** *Under the same conditions as Proposition 10, fix any $T > 0$. For any $t \in (0, T]$ let $K^p_{\phi(t)} : \mathcal{X} \times \mathcal{X} \to \mathbb{R}^{q \times q}$ be the mapping defined by $(x, x') \mapsto K_{\phi(t)}(x, x') = J_\phi f(x; \phi(t)) J_\phi f(x'; \phi(t))^\top$. Then provided conditions (D1)–(D4) hold, we have as $p \to \infty$ that*

$$\sup_{x, \tilde{x} \in \mathcal{X}, t \in (0, T]} ||K^p_{\phi(t)}(x, \tilde{x}) - K^p_{\phi(0)}(x, \tilde{x})||_F \overset{a.s.}{\to} 0. \tag{LT}$$

**Proof** Let us examine the $k, l$th term of the $q \times q$ matrix given by $K^p_{\phi(t)}(x, \tilde{x}) - K^p_{\phi(0)}(x, \tilde{x})$ for fixed $x, \tilde{x}$, and some $t \in (0, T]$. The $k, l$th term is given by (see the work in Appendix C):

$$\frac{1}{p} \sum_{j=1}^{p} \mathbf{1}_{k=l} \left( \sigma \left( x^\top w_j(t) \right) \sigma \left( \tilde{x}^\top w_j(t) \right) \right) - \tag{15}$$

$$\left( \sigma \left( x^\top w_j(0) \right) \sigma \left( \tilde{x}^\top w_j(0) \right) \right) \tag{16}$$

$$+ \frac{1}{p} \sum_{j=1}^{p} \left( a_{kj}(t) a_{lj}(t) \sigma' \left( x^\top w_j(t) \right) \sigma' \left( \tilde{x}^\top w_j(t) \right) \left( x^\top \tilde{x} \right) \right) - \tag{17}$$

$$\left( a_{kj}(0) a_{lj}(0) \sigma' \left( x^\top w_j(0) \right) \sigma' \left( \tilde{x}^\top w_j(0) \right) \left( x^\top \tilde{x} \right) \right). \tag{18}$$

Above, we have explicitly made clear the dependence of the parameters on time, e.g. $w_j(t)$ vs. $w_j(0)$. We aim to show that the quantity above tends to zero as $p \to \infty$. We first prove this holds pointwise, and will consider the red and blue terms one at a time for a fixed $x, \tilde{x}$.

First consider the $j$th summand of the red term. We will bound its absolute value. If $k \neq l$, we're done, so assume $k = l$. We have for any $j$ that

$$\left| \sigma \left( x^\top w_j(t) \right) \sigma \left( \tilde{x}^\top w_j(t) \right) - \sigma \left( x^\top w_j(0) \right) \sigma \left( \tilde{x}^\top w_j(0) \right) \right|$$

$$= \left| \sigma \left( x^\top w_j(t) \right) \sigma \left( \tilde{x}^\top w_j(t) \right) - \sigma \left( x^\top w_j(t) \right) \sigma \left( \tilde{x}^\top w_j(0) \right) + \right.$$

$$\left. \sigma \left( x^\top w_j(t) \right) \sigma \left( \tilde{x}^\top w_j(0) \right) - \sigma \left( x^\top w_j(0) \right) \sigma \left( \tilde{x}^\top w_j(0) \right) \right|$$

$$\leq |\sigma \left( x^\top w_j(t) \right)| \cdot |\sigma \left( \tilde{x}^\top w_j(t) \right) - \sigma \left( \tilde{x}^\top w_j(0) \right)| + |\sigma \left( \tilde{x}^\top w_j(0) \right)| \cdot |\sigma \left( x^\top w_j(t) \right) - \sigma \left( x^\top w_j(0) \right)|$$

and by the Lipschitz assumption on $\sigma(\cdot)$ and Cauchy-Schwarz, we can bound the quantity above as follows

$$\leq (K + C||x||_2 ||w_j(t)||_2) \cdot C||\tilde{x}||_2 ||w_j(t) - w_j(0)||_2 + (K + C||x||_2 ||w_j(0)||_2) \cdot C||x||_2 ||w_j(t) - w_j(0)||_2$$

$$= C^2 ||w_j(t) - w_j(0)||_2 \left( 2\frac{K}{C} + ||w_j(t)||_2 + ||w_j(0)||_2 \right) \quad \text{since } ||x||_2, ||\tilde{x}||_2 \text{ are bounded by (D1)}$$

$$\leq C^2 ||w_j(t) - w_j(0)||_2 \left( 2\frac{K}{C} + ||w_j(t) - w_j(0)||_2 + 2||w_j(0)||_2 \right) \quad \text{by triangle inequality}$$

$$= 2CK ||w_j(t) - w_j(0)||_2 + C^2 ||w_j(t) - w_j(0)||_2^2 + 2C^2 ||w_j(0)||_2 ||w_j(t) - w_j(0)||_2$$

and using Lemma 11, we can bound all terms above using $||w_j(0)||_2$ as follows.

$$\leq 2CK\left(D_{p,t}||w_j(0)||_2 + E_{p,t}\right) + C^2\left(D_{p,t}||w_j(0)||_2 + E_{p,t}\right)^2 + 2C^2\left(D_{p,t}||w_j(0)||_2^2 + E_{p,t}||w_j(0)||_2\right)$$
$$= \left(2C^2 D_{p,t} + C^2 D_{p,t}^2\right)||w_j(0)||_2^2 + \left(2CKD_{p,t} + 2C^2 D_{p,t}E_{p,t} + 2C^2 E_{p,t}\right)||w_j(0)||_2 + \left(2CKE_{p,t} + C^2 E_{p,t}^2\right)$$

Recalling that $w_j(0) \overset{iid}{\sim} \mathcal{N}(0, I_d)$, we have that $||w_j(0)||_2$ and $||w_j(0)||_2^2$ are integrable with expectations denoted $\mu$ and $\nu$, respectively. All our work has allowed us to show that

$$\left|\frac{1}{p}\sum_{j=1}^{p}\left(\sigma\left(x^\top w_j(t)\right)\sigma\left(\tilde{x}^\top w_j(t)\right) - \sigma\left(x^\top w_j(0)\right)\sigma\left(\tilde{x}^\top w_j(0)\right)\right)\right|$$
$$\leq \frac{1}{p}\sum_{j=1}^{p}\left(2C^2 D_{p,t} + C^2 D_{p,t}^2\right)||w_j(0)||_2^2 + \left(2CKD_{p,t} + 2C^2 D_{p,t}E_{p,t} + 2^2 CE_{p,t}\right)||w_j(0)||_2$$
$$+ \left(2CKE_{p,t} + C^2 E_{p,t}^2\right)$$
$$\overset{a.s.}{\to}\left(\lim_{p\to\infty} 2C^2 D_{p,t} + C^2 D_{p,t}^2\right)\nu + \left(\lim_{p\to\infty} 2CKD_{p,t} + 2C^2 D_{p,t}E_{p,t} + 2C^2 E_{p,t}\right)\mu$$
$$+ \left(\lim_{p\to\infty} 2CKE_{p,t} + C^2 E_{p,t}^2\right)$$
$$= 0$$

by conditions on $D_{p,t}$ and $E_{p,t}$ from Lemma 11, the strong law of large numbers, and the classic result from analysis that $\lim_{n\to\infty} a_n b_n = \left(\lim_{n\to\infty} a_n\right)\left(\lim_{n\to\infty} b_n\right)$, provided both limits on the right hand side exist. Lastly, we can achieve the same result for the blue term quickly. Because $a_{ij}(0) = 0$ w.p. 1 by (D2), we have almost surely that

$$\frac{1}{p}\sum_{j=1}^{p}\left(a_{kj}(t)a_{lj}(t)\sigma'\left(x^\top w_j(t)\right)\sigma'\left(\tilde{x}^\top w_j(t)\right)\left(x^\top\tilde{x}\right)\right) -$$
$$\cancel{\left(a_{kj}(0)a_{lj}(0)\sigma'\left(x^\top w_j(0)\right)\sigma'\left(\tilde{x}^\top w_j(0)\right)\left(x^\top\tilde{x}\right)\right)}$$
$$\leq \frac{1}{p}\sum_{j=1}^{p}|a_{kj}(t)||a_{lj}(t)||\sigma'\left(x^\top w_j(0)\right)||\sigma'\left(\tilde{x}^\top w_j(0)\right)|||x||_2||\tilde{x}||_2$$
$$\leq \frac{C^2}{p}\sum_{j=1}^{p}|a_{kj}(t)||a_{lj}(t)|$$
$$\leq \frac{C^2}{p}\sum_{j=1}^{p}||a_j(t)||_2^2$$

because for all $j$, we have $|a_{kj}|, |a_{lj}|$ are dominated by $||a_j||_2$. From here, we have by Lemma 12 that we can bound each term in the sum above by

$$\leq \frac{C^2}{p} \sum_{j=1}^{p} (||w_j(0)||_2 F_{p,t} + G_{p,t})^2$$

$$= \frac{C^2}{p} \sum_{j=1}^{p} F_{p,t}^2 ||w_j(0)||_2^2 + 2F_{p,t}G_{p,t}||w_j(0)||_2 + G_{p,t}^2$$

$$\overset{a.s.}{\to} 0$$

as $p \to \infty$ by similar logic to the above. Together, these results combine to show that $|K^p_{\phi(t)}(x, \tilde{x})_{kl} - K^p_{\phi(0)}(x, \tilde{x})_{kl}| \overset{a.s.}{\to} 0$ as $p \to \infty$. As $k, l$ were arbitrary $k, l \in 1, \ldots, q$, we have $||K^p_{\phi(t)}(x, \tilde{x}) - K^p_{\phi(0)}(x, \tilde{x})||_F \overset{a.s.}{\to} 0$. This establishes pointwise convergence for some fixed $t \in (0, T]$. Uniform convergence over all of $\mathcal{X} \times \mathcal{X}$ and all $t \in (0, T]$ follows easily in this case. Firstly, the numbers $D_{p,t}, E_{p,t}, F_{p,t}$, and $G_{p,t}$ are monotonic in $t$, so we can bound uniformly for all $t \in (0, T]$ by taking $t = T$ in the expressions above. Secondly, in our work above, our bounds on the red and blue terms were independent of the choice of point $(x, \tilde{x})$. More precisely, the supremum over $x, \tilde{x}$ can accounted for in the bounds easily by observing that $\sup_{x,\tilde{x} \in \mathcal{X}, t \in (0,T]} ||K^p_{\phi(t)}(x, \tilde{x}) - K^p_{\phi(0)}(x, \tilde{x})||_F$ can be bounded above by

$$\leq \sup_{x,\tilde{x} \in \mathcal{X}} \left\| \frac{1}{p} \sum_{j=1}^{p} \mathbf{1}_{k=l} \left( \sigma\left(x^\top w_j(t)\right) \sigma\left(\tilde{x}^\top w_j(t)\right) \right) \right.$$

$$\left. - \left( \sigma\left(x^\top w_j(0)\right) \sigma\left(\tilde{x}^\top w_j(0)\right) \right) \right\|$$

$$+ \sup_{x,\tilde{x} \in \mathcal{X}} \left\| \frac{1}{p} \sum_{j=1}^{p} \left( a_{kj}(t)a_{lj}(t)\sigma'\left(x^\top w_j(t)\right) \sigma'\left(\tilde{x}^\top w_j(t)\right) \left(x^\top \tilde{x}\right) \right) \right.$$

$$\left. - \left( a_{kj}(0)a_{lj}(0)\sigma'\left(x^\top w_j(0)\right) \sigma'\left(\tilde{x}^\top w_j(0)\right) \left(x^\top \tilde{x}\right) \right) \right\|$$

$$\leq \sup_{x,\tilde{x} \in \mathcal{X}} \frac{1}{p} \sum_{j=1}^{p} \left(2C^2 D_{p,T} + C^2 D_{p,T}^2\right) ||w_j(0)||_2^2 + \left(2CKD_{p,t} + 2C^2 D_{p,T} E_{p,T} + 2C^2 E_{p,T}\right) ||w_j(0)||_2$$

$$+ \left(2CKE_{p,T} + C^2 E_{p,T}^2\right)$$

$$+ \sup_{x,\tilde{x} \in \mathcal{X}} \frac{C^2}{p} \sum_{j=1}^{p} F_{p,T}^2 ||w_j(0)||_2^2 + 2F_{p,T}G_{p,T}||w_j(0)||_2 + G_{p,T}^2$$

$$\overset{a.s.}{\to} 0$$

by the same work as above. ∎

## Appendix E. Kernel Gradient Flow Analysis

We rely on additional regularity conditions outlined below. We will consider the following three flows in our proof of Theorem 4 (for some choice of $p$):

$$\dot{f}_t(x) = -\mathbb{E}_{P(X)} K^p_{\phi(t)}(x, X) \ell'(X, f_t(X)) \tag{19}$$

$$\dot{g}_t(x) = -\mathbb{E}_{P(X)} K_\infty(x, X) \ell'(X, g_t(X)) \tag{20}$$

$$\dot{h}_t(x) = -\mathbb{E}_{P(X)} K^p_{\phi(0)}(x, X) \ell'(X, h_t(X)) \tag{21}$$

where $f_t$ is shorthand for $f(\cdot; \phi(t))$. The three flows above can thought of as corresponding to $L_P$, $L_F$, and a "lazy" variant, respectively. The flow of $h_t$ is "lazy" because it follows the dynamics of a fixed kernel, the kernel at initialization. The flow of $g_t$ also follows a fixed kernel, but the limiting NTK $K_\infty$ instead. The flow of $f_t$ is that obtained in practice, where the kernel $K^p_{\phi(t)}$ changes continuously as the parameters $\phi(t)$ evolve in time. The flow in $h_t$ is used to bound the differences between $f_t$ and $g_t$ in the proof of Theorem 4. We now enumerate the regularity conditions.

(E1) The functional $L_F(f)$ satisfies $\inf_f L_F(f) > -\infty$.

(E2) The limiting NTK $K_\infty$ is positive definite (so that the RKHS $\mathcal{H}$ with kernel $K_\infty$ is well-defined).

(E3) Under (E1) and (E2), the function $f^*$ minimizing $L_F$ satisfies $||f^*||_{\mathcal{H}} < \infty$, so that $f^* \in \mathcal{H}$.

(E4) For any choice of $p$, we have for all $t, x$ that $\ell'(x; f_t(x)), \ell'(x; g_t(x))$, and $\ell'(x; h_t(x))$ are bounded by a constant $\tilde{M}$.

(E5) The function $\ell$ is $\tilde{L}$-smooth in its second argument, i.e. $||\ell'(x, \eta_1) - \ell'(x, \eta_2)|| \leq \tilde{L}||\eta_1 - \eta_2||$.

We first prove Lemma 3 from the manuscript.

**Lemma 3** *Let $f^*$ denote the minimizer of $L_F$ from Lemma 1, and $\epsilon > 0$. Fix $f_0$, and let $K_\infty$ denote the limiting neural tangent kernel. Let $f_0$ evolve according to the dynamics*

$$\dot{f}_t(x) = -\mathbb{E}_{P(X)} K_\infty(x, X) \ell'(X, f_t(X)).$$

*Suppose that the conditions of Lemma 1 and (E1)-(E3) hold. Then, there exists $T > 0$ such that $L_F(f_T) \leq L_F(f^*) + \epsilon$.*

**Proof** Let $f^* \in \operatorname{argmin} L_F(f)$, where $L_F(f)$ is the functional objective. Then $L_F(f^*) > -\infty$ by (E1). Hereafter, let $L_F = L$ for notational convenience. Then if $f_t$ evolves according to the kernel gradient flow above, we have (from the chain rule for Fréchet derivatives) that

$$\dot{L}(f_t) = \frac{\partial L}{\partial f_t} \circ \frac{\partial f_t}{\partial t}.$$

By definition, $\frac{\partial f_t}{\partial t} = \dot{f}_t$. We also have $\frac{\partial L}{\partial f_t} : h \mapsto \mathbb{E}_{P(X)} \ell'(X, f_t(X))^\top h(X)$. Plugging this in yields

$$\dot{L}(f_t) = \mathbb{E}_{X \sim P(X)} \ell'(X, f_t(X))^\top \left[ -\mathbb{E}_{X' \sim P(X)} K_\infty(X, X') \ell'(X', f_t(X')) \right]$$
$$= -\mathbb{E}_{X, X' \sim P} \ell'(X, f_t(X))^\top K_\infty(X, X') \ell'(X', f_t(X')) \leq 0$$

by the positiveness of the kernel $K_\infty$ (from (E2)). Now define $\Delta_t = \frac{1}{2} ||f_t - f^*||_{\mathcal{H}}^2$, where $\mathcal{H}$ is the vector-valued reproducing kernel Hilbert space corresponding to the kernel $K_\infty$ (see Carmeli et al. (2006) for a detailed review). It follows that $\frac{\partial \Delta_t}{\partial f_t} : h \mapsto \langle f_t - f^*, h \rangle$. Then by the chain rule we have

$$-\dot{\Delta}_t = -\langle f_t - f^*, \dot{f}_t \rangle$$
$$= -\langle f_t - f^*, -\mathbb{E}_{P(X)} K_\infty(\cdot, X) \ell'(X, f_t(X)) \rangle$$
$$= \mathbb{E}_{P(X)} \ell'(X, f_t(X))^\top [f_t(X) - f^*(X)]$$
$$\geq \mathbb{E}_{P(X)} \ell(X, f_t(X)) - \ell(X, f^*(X))$$
$$= L(f_t) - L(f^*).$$

To go from the second to the third line, we used the reproducing property of the vector-valued kernel, the definition of inner product, and the linearity of integration. More precisely, the reproducing property (cf. Eq. (2.2) of Carmeli et al. (2006)) tells us for any functions $g, h$ and fixed $x$,

$$\langle g, K_\infty(\cdot, x) h(x) \rangle = g(x)^\top h(x)$$

and so the third line results from the second by exchanging the integral and inner product. In the second-to-last line we used convexity of $\ell$ in its second argument (from Lemma 1 of the manuscript). Now consider the Lyapunov functional given by

$$\mathcal{E}(t) = t \left[ L(f_t) - L(f^*) \right] + \Delta_t. \tag{22}$$

Differentiating, we have

$$\dot{\mathcal{E}}(t) = L(f_t) - L(f^*) + t\dot{L}(f_t) + \dot{\Delta}_t \leq 0$$

by the above work because i) $t\dot{L}(f_t) \leq 0$ and ii) $L(f_t) - L(f^*) + \dot{\Delta}_t \leq 0$, implying that $\mathcal{E}(t) \leq \mathcal{E}(0)$ for all $t$. Evaluating at $t = 0$, thus

$$t \left[ L(f_t) - L(f^*) \right] + \Delta_t \leq \Delta_0$$
$$t \left[ L(f_t) - L(f^*) \right] \leq \Delta_0 - \Delta_t$$
$$t \left[ L(f_t) - L(f^*) \right] \leq \Delta_0 \quad \text{since } \Delta_t \geq 0$$
$$\left[ L(f_t) - L(f^*) \right] \leq \frac{1}{t} \Delta_0.$$

and so we have that there exists sufficiently large $T$ such that $|L(f_T) - L(f^*)| \leq \epsilon$ as desired. ∎

Using this result and our previous results, we now are able to prove Theorem 4 from the manuscript.

**Theorem 4** *Consider the width-p scaled 2-layer ReLU network, evolving via the flow*

$$\dot{f}_t(x) = -\mathbb{E}_{P(X)} K^p_{\phi(t)}(x, X)\ell'(X, f_t(X)), \tag{5}$$

*where $f_t$ denotes $f(\cdot, \phi(t))$. Let $f^*$ denote the unique minimizer of $L_F$ from Lemma 1, and fix $\epsilon > 0$. Then, under conditions (C1)–(C4),(D1)–(D4), and (E1)–(E5), there exists $T > 0$ such that almost surely*

$$\left[\lim_{p \to \infty} L(f_T)\right] \leq L(f^*) + \epsilon, \tag{6}$$

*where $L$ is the loss functional $L_F$.*

**Proof** Throughout, we use $L$ to denote $L_F$ for ease. We will examine the three gradient flows

$$\dot{f}_t(x) = -\mathbb{E}_{P(X)} K^p_{\phi(t)}(x, X)\ell'(X, f_t(X)) \tag{23}$$

$$\dot{g}_t(x) = -\mathbb{E}_{P(X)} K_\infty(x, X)\ell'(X, g_t(X)) \tag{24}$$

$$\dot{h}_t(x) = -\mathbb{E}_{P(X)} K^p_{\phi(0)}(x, X)\ell'(X, h_t(X)) \tag{25}$$

and establish the result by the triangle inequality, i.e.

$$|L(f_T) - L(f^*)| \leq |L(f_T) - L(g_T)| + |L(g_T) - L(f^*)|. \tag{26}$$

The flow in $h_t$ will be used to help bound the first term, but we begin with the second term. By Lemma 3, pick $T > 0$ sufficiently large such that $|L(g_T) - L(f^*)| \leq \epsilon/2$. Fix this $T$. This provides a suitable bound on the second term.

Turning to the first term, by continuity of $L(f)$ in $f$, there exists $\delta > 0$ such that $y \in B(g_T, \delta) \implies |L(y) - L(g_T)| \leq \epsilon/2$. We will show that there exists $P$ sufficiently large such that $p > P$ implies $||f_T - g_T|| \leq \delta$ almost surely, yielding the desired bound on the first term of the decomposition above. Throughout, $||\cdot||$ denotes the $L^2$ norm of a function with respect to probability measure $P(X)$ (i.e. the marginal distribution of our joint model $P(\Theta, X)$).

To show that there exists sufficiently large $P$ such that $||f_T - g_T|| \leq \delta$, we use another application of the triangle inequality

$$||f_T - g_T|| \leq ||f_T - h_T|| + ||h_T - g_T||$$

and construct bounds on the two terms on the right hand side using Proposition 10 and Proposition 13. Observe first that by (C2)/(D2), at initialization we have almost surely that $f_0 = g_0 = h_0 = 0$. Also note that by continuity of $K_\infty$ (established in Lemma 9) on the compact domain $\mathcal{X} \times \mathcal{X}$ we have $\sup_{x,\tilde{x}} ||K_\infty(x, \tilde{x})||_2 < M$ for some $M$. Finally, note that by (E5) the function $\ell'(x, \eta)$ is Lipschitz in its second argument with constant $\tilde{L}$. Below, we let $||\cdot||_2$ denote the 2-norm of a vector or matrix, depending on the argument, and $||\cdot||_F$ the Frobenius norm of a matrix. For functions, as stated $||\cdot||$ denotes the $L^2$

norm with respect to measure $P(X)$, i.e. $||f||^2 = \int f(X)^\top f(X) dP(X)$. From here, we have

$$
\begin{aligned}
||g_T - h_T|| &\overset{a.s.}{=} ||(g_T - g_0) - (h_T - h_0)|| \\
&= \left\| \int_0^T \mathbb{E}_{P(X)} \left[ K_\infty(\cdot, X)\ell'(X, g_t(X)) - K_{\phi(0)}^p(\cdot, X)\ell'(X, h_t(X)) \right] dt \right\| \\
&\leq \int_0^T \mathbb{E}_{P(X)} ||K_\infty(\cdot, X)\ell'(X, g_t(X)) - K_{\phi(0)}^p(\cdot, X)\ell'(X, h_t(X))|| dt \\
&= \int_0^T \mathbb{E}_{P(X)} ||K_\infty(\cdot, X)\ell'(X, g_t(X)) - K_\infty(\cdot, X)\ell'(X, h_t(X)) + \\
&\qquad K_\infty(\cdot, X)\ell'(X, h_t(X)) - K_{\phi(0)}^p(\cdot, X)\ell'(X, h_t(X))|| dt \\
&\leq \int_0^T \mathbb{E}_{P(X)} ||K_\infty(\cdot, X) \left[ \ell'(X, g_t(X)) - \ell'(X, h_t(X)) \right] || + \\
&\qquad ||K_\infty(\cdot, X)\ell'(X, h_t(X)) - K_{\phi(0)}^p(\cdot, X)\ell'(X, h_t(X))|| dt \qquad (27)
\end{aligned}
$$

Now, we note the following facts. Firstly, for any kernel $K$ that is uniformly bounded (i.e. $||K(x,y)||_2 \leq M$ for any $x, y$), the $L^2$ norm of the function $||K(\cdot, X)v||$ for fixed $X, v$ can be bounded in terms of $M$ and $||v||_2$ because

$$
||K(\cdot, X)v||^2 = \int v^\top K(Y, X)^\top K(Y, X) v dP(Y) \leq \int ||K(Y, X)||_2^2 ||v||_2^2 dP(Y) \leq M^2 ||v||_2^2
$$
$$
\implies ||K(\cdot, X)v|| \leq M ||v||_2.
$$

Secondly, we have again for any fixed $v$ and $X$ that

$$
\begin{aligned}
\left\| \left[ K_\infty(\cdot, X) - K_{\phi(0)}^p(\cdot, X) \right] v \right\|^2 &= \int v^\top \left[ K_\infty(Y, X) - K_{\phi(0)}^p(Y, X) \right]^\top \left[ K_\infty(Y, X) - K_{\phi(0)}^p(Y, X) \right] v dP(y) \\
&\leq \int ||K_\infty(Y, X) - K_{\phi(0)}^p(Y, X)||_2^2 ||v||_2^2 dP(y) \\
&\leq \left( \sup_{x,y} ||K_\infty(y, x) - K_{\phi(0)}^p(y, x)||_F \right)^2 ||v||_2^2 \\
\implies \left\| \left[ K_\infty(\cdot, X) - K_{\phi(0)}^p(\cdot, X) \right] v \right\| &\leq \sup_{x,y} ||K_\infty(y, x) - K_{\phi(0)}^p(y, x)||_F \cdot ||v||_2
\end{aligned}
$$

since the matrix (spectral) 2-norm is dominated by the Frobenius norm. Plugging these facts into Equation (27) above, we have

$$
\leq \int_0^T \mathbb{E}_{P(X)} M \cdot ||\ell'(X, g_t(X)) - \ell'(X, h_t(X))||_2 + \sup_{x,y} ||K_\infty(x,y) - K^p_{\phi(0)}(x,y)||_F \cdot ||\ell'(X, h_t(X))||_2 dt
$$

$$
\leq \int_0^T \mathbb{E}_{P(X)} M\tilde{L} ||g_t(X) - h_t(X)||_2 + \tilde{M} \sup_{x,y} ||K_\infty(x,y) - K^p_{\phi(0)}(x,y)|| dt \text{ by (E4)}
$$

$$
\leq \tilde{M}T \sup_{x,y} ||K_\infty(x,y) - K^p_{\phi(0)}(x,y)||_F + M\tilde{L} \int_0^T \mathbb{E}_{P(X)} \sqrt{||g_t(X) - h_t(X)||_2^2} dt
$$

$$
\leq \tilde{M}T \sup_{x,y} ||K_\infty(x,y) - K^p_{\phi(0)}(x,y)||_F + M\tilde{L} \int_0^T \sqrt{\mathbb{E}_{P(X)} ||g_t(X) - h_t(X)||_2^2} dt \text{ by Jensen's inequality}
$$

$$
= \tilde{M}T \sup_{x,y} ||K_\infty(x,y) - K^p_{\phi(0)}(x,y)||_F + M\tilde{L} \int_0^T ||g_t - h_t|| dt
$$

and so by Gronwall's inequality (Theorem 7), we have almost surely that

$$
||g_T - h_T|| \leq \tilde{M}T \sup_{x,y} ||K_\infty(x,y) - K^p_{\phi(0)}(x,y)||_F \exp(M\tilde{L}T).
$$

By Proposition 10, there thus exists $P_1$ such that for all $p > P_1$ we have $||g_T - h_T|| \leq \frac{\delta}{2}$ almost surely. We proceed nearly identically for the term $||h_T - f_T||$. We need only note that for sufficiently large $p$, say $p > P_2$, we can bound $K^p_{\phi(0)}$ uniformly (almost surely) by a constant $A > M$. To see this, observe that by Proposition 10 we have that there exists almost surely a sufficiently large $P$ such that $p > P$ implies $\sup_{x,y} ||K_\infty(x,y) - K^p_{\phi(0)}(x,y)||_F < A - M$ and so by triangle inequality we have for all $p > P$ almost surely that

$$
\sup_{x,y} ||K^p_{\phi(0)}||_F \leq \sup_{x,y} ||K^p_{\phi(0)}(x,y) - K_\infty(x,y)||_F + ||K_\infty(x,y)||_F
$$

$$
\leq \sup_{x,y} ||K^p_{\phi(0)}(x,y) - K_\infty(x,y)||_F + \sup_{x,y} ||K_\infty(x,y)||_F
$$

$$
\leq A - M + M = A.
$$

Thereafter,

$$
\begin{aligned}
||h_T - f_T|| &\overset{a.s.}{=} ||(h_T - h_0) - (f_T - f_0)|| \\
&= \left|\left| \int_0^T \mathbb{E}_{P(X)} \left[ K_{\phi(0)}^p(\cdot, X)\ell'(X, h_t(X)) - K_{\phi(t)}^p(\cdot, X)\ell'(X, f_t(X)) \right] dt \right|\right| \\
&\leq \int_0^T \mathbb{E}_{P(X)} ||K_{\phi(0)}^p(\cdot, X)\ell'(X, h_t(X)) - K_{\phi(t)}^p(\cdot, X)\ell'(X, f_t(X))||dt \\
&\leq \int_0^T \mathbb{E}_{P(X)} ||K_{\phi(0)}^p(\cdot, X)\ell'(X, h_t(X)) - K_{\phi(0)}^p(\cdot, X)\ell'(X, f_t(X))|| + \\
&\qquad ||K_{\phi(0)}^p(\cdot, X)\ell'(X, f_t(X)) - K_{\phi(t)}^p(\cdot, X)\ell'(X, f_t(X))||dt \\
&\leq \int_0^T \mathbb{E}_{P(X)} A \cdot ||\ell'(X, h_t(X)) - \ell'(X, f_t(X))||_2 + \\
&\qquad \sup_{x,y,t\in(0,T]} ||K_{\phi(0)}^p(x,y) - K_{\phi(t)}^p(x,y)||_F \cdot ||\ell'(X, f_t(X))||dt \\
&\leq \tilde{M}T \sup_{x,y,t\in(0,T]} ||K_{\phi(0)}^p(x,y) - K_{\phi(t)}^p(x,y)||_F + A\tilde{L} \int_0^T \mathbb{E}_{P(X)} ||h_t(X) - f_t(X)||_2 dt
\end{aligned}
$$

and we can similarly switch from $\mathbb{E}_{P(X)} ||h_t(X) - f_t(X)||_2$ to the $L_2$ norm $||h_t - f_t||$ as above using Jensen's inequality, yielding

$$
\begin{aligned}
&\leq \tilde{M}T \sup_{x,y,t\in(0,T]} ||K_{\phi(0)}^p(x,y) - K_{\phi(t)}^p(x,y)||_F + A\tilde{L} \int_0^T ||h_t - f_t||dt \\
&\implies ||h_T - f_T|| \leq \tilde{M}T \sup_{x,y,t\in(0,T]} ||K_{\phi(0)}^p(x,y) - K_{\phi(t)}^p(x,y)||_F \exp\left( A\tilde{L}T \right)
\end{aligned}
$$

almost surely. Clearly, by the same logic as the above there exists $P_3$ such that $p > P_3$ implies $\tilde{M}T \sup_{x,y,t\in(0,T]} ||K_{\phi(0)}^p(x,y) - K_{\phi(0)}^p(x,y)|| \exp(A\tilde{L}T) \leq \delta/2$ by Proposition 13. Then for all $p > \max(P_1, P_2, P_3)$, we have almost surely that $||h_T - f_T|| \leq \delta/2$. This completes the proof, as in this case we have by the triangle inequality that $||f_T - g_T|| \leq \delta$ and so $|L(f_T) - L(g_T)| \leq \epsilon/2$ by construction. ∎

## Appendix F. Experimental Details

We use PyTorch (Paszke et al., 2019) to implement our experiments[1], with permission of the license. All experiments utilized NVIDIA GeForce RTX 2080 Ti graphical processing units (GPUs), and fit within 10 GB of memory.

Recall the generative model for this problem, given by the following:

$$
\begin{aligned}
\Theta &\sim \text{Unif}[0, 2\pi] \\
Z &\sim \mathcal{N}(0, \sigma^2) \\
X \mid (\Theta = \theta, Z = z) &\sim \delta\left( [\cos(\theta + z), \sin(\theta + z)]^\top \right).
\end{aligned}
$$

---

1. Code publicly available at https://github.com/declanmcnamara/gcvi_aabi.

The variable $\sigma$ is a hyperparameter of the model that we take to be $\sigma = 0.5$. The model is constructed such that $x \in \mathbb{S}^1$ to satisfy assumptions (C1) and (D1), respectively. One thousand pairs of data points $\{\theta_i, x_i\}_{i=1}^{1000}$ were generated independently from the model above and fixed as the "dataset" for which ground truth latent parameter values are known.

We constructed scaled, dense single hidden-layer ReLU networks of varying widths, with $2^j$ neurons for $j = 6, \ldots, 12$ with the same architecture as in Appendix C and the initialization described in condition (C2). All networks were trained to minimize the expected forward KL objective $L_P$; stochastic gradients were estimated using batches of 16 independent simulated $(\theta, x)$ pairs from the generative model, and stochastic gradient descent was performed using the Adam optimizer with learning rate $\rho = 0.0001$. We employ a learning rate scheduler that scales the learning rate as $O(1/I)$, where $I$ denotes the number of iterations. All models were fitted for 200,000 stochastic gradient steps, and execution time is less than one hour. The natural parameter for the von Mises distribution is parameterized as $\eta = f(x; \phi) + 0.0001$. This small perturbation must be added because $f(\cdot; \phi) = 0$ at initialization, and the value of $\eta = 0$ lies outside the natural parameter space for this variational family.

For the linearized neural network models, all training settings where the same except for the architecture. For these models, we first constructed neural networks as above for each width to compute the Jacobian evaluated at the initial weights $J_\phi(x; \phi_0)$. Thereafter, the model in $\phi$ is fixed as

$$f(x; \phi) = f(x; \phi_0) + J_\phi(x; \phi_0)(\phi - \phi_0)$$

where $\phi, \phi_0$ are flattened vectors of parameters from the neural network architectures. Using this linearized model above, the parameter $\phi$ is fitted by SGD as above.

The plots in Figure 1 of the manuscript are constructed by evaluating the average negative log-likelihood on the dataset at each iteration, i.e. for the fixed $n = 1000$ pairs of observations above, we evaluate the finite-sample loss for the expected forward KL divergence. Up to fixed constants, this quantity is given by

$$-\frac{1}{n} \sum_{i=1}^{n} \log q(\theta_i; f(x_i; \phi))$$

where $\phi$ is the current iterate of the parameters (either the neural network parameters or the flattened vector of parameters of the same size for the linearized model). The red horizontal line in Figure 1 is set at the value $-\frac{1}{n} \sum_{i=1}^{n} \log p(\theta_i \mid x_i)$, where $p$ denotes the exact posterior distribution (computed using numerical integration over a fine grid of evaluation points).

