# OpenReview forum: "Globally Convergent Variational Inference"
_approximateinference.org/AABI/2024/Symposium — AABI 2024_

### Official Review · Reviewer_sgjK · 2024-04-22

**Rating:** 6
**Confidence:** 2

**Review:**

I must say upfront, I am not really fully equipped to provide a detailed review of this paper.  I am unfamiliar with NTKs and so I cannot evaluate what I feel is the most important punchline of the paper.

**Strengths:**

This paper presents an analysis of a functional form of the expected forward KL divergence:  an appealing alternative to the classic ELBO loss used in variational inference.  I believe the core message [and I invite the authors to correct me] is that the expected forward KL optimization objective, when parameterized with exponential families, is strictly convex;  and hence poses a well-behaved optimization problem.  This makes a lot of sense, and is a nice, if not entirely surprising, takeaway.

**Weaknesses:**

From my understanding, the core result is that there is a unique exponential family maximizer to the objective.  However, as soon as you parameterize this with a neural network, the objective is no longer convex in the parameters you can actually optimize -- its fine being convex in a function, but you can't actually optimize the function, you can only optimize the parameters, and then it becomes non-convex.  I therefore struggle to see the point of making this distinction.

I also struggle to keep up with with the thread of the paper, completely losing it by Section 4 & 5.  I don't think the current presentation does an especially good job of relating, within the prose, each thought to the overarching objectives.  This makes the paper very hard to digest and situate the threads.  There is no abstract, conclusion, summary prior to Section 5, or explanation of what the experiment is trying to validate and why.  This really harms the perceived clarity and significance of the work, as the reader (at least me) is left without a coherent message.

Furthermore, the experiment doesn't actually validate (as far as I can tell) that the objective provides a tractable or performant method for learning models?  I was expecting to see comparisons to the ELBO objective on a synthetic problem with a closed form or MCMC-derived posterior.

Generally, I think the paper itself is prepared fairly well;  but does very little to aid the reader.  The information is laid out almost in a narrative form;  instead of collecting and summarizing the key ideas, and then digging into the detail elsewhere.  The authors are to be commended for the detail in the appendix, but that much information actually makes it impenetrable, and is a hinderance instead of a help.

**Summary:**

Overall I think this paper is okay.  I struggle to follow the thread and punchline, and I am not proficient enough with NTKs to offer guidance on that aspect of the paper.  Altogether, this makes it difficult for me to really positively endorse the paper as it stands.  I am willing to change my evaluation though if the authors can convince me otherwise.

Good work, and good luck.

---

### Official Review · Reviewer_wHEx · 2024-04-24
**Interesting work, but misleading title and no abstract.**

**Rating:** 4
**Confidence:** 2

**Review:**

Reviewing this paper was a bit of a strange experience, because the core content of the paper appears as interesting and high-quality work, but the title and form seem inappropriate to me for several reasons:

- The title: I think the title of the paper is simply misleading. "Globally convergent variational inference" sounds very general and suggests (at least to me) a general approach for making variational inference globally convergent. While I agree that the work of the paper falls in the category of variational inference, it is a very, very specialized form of variational inference and the results (as I see them) are not really applicable outside this specialized setting. Therefore, I think the title needs to be much more specific.

- The length of the paper: The length of the paper is 6 pages + 20 pages of appendices. However, the main paper is insufficient in several ways: 1) the paper is missing an abstract and it would violate the page limit if an abstract was added, and hence, the paper is too long, 2) the specific objective in eq. (1) could be better motivated, 3) there is very little discussion of the significance of the results or why it is interesting

I did not verify the proofs in the appendix and therefore, I do not have strong opinions on the theoretical results. Hence, my score
is mostly based on the arguments above.

Other comments:
- "We examine an increasingly popular alternative objective for variational inference, the forward KL divergence." Perhaps provide a couple of references to justify this?

- The motivation and key take-away for the numerical experiment could be better described.

- Please correct me if I am wrong, but it seems to me that the Lemma 1 is the standard result that the solution for q^ = \arg\min_q KL[p||q] for q in the exponential family is equal to moment matching. If so, this could reduce the length of paper and make room for an abstract and more elaborate explanations.

---

### Official Review · Reviewer_hasj · 2024-04-25
**A convergence guarantee for VI parametrized with a neural network**

**Rating:** 7
**Confidence:** 3

**Review:**

In this paper authors show, that typically highly non-convex problem of minimizing expected KL divergence, can reach a global minimum under certain conditions. Authors focus on amortized inference problem, and parametrize the variational approximation with a neural network. If the variational approximation belongs to the exponential family, authors show that the forward KL objective is strictly convex with respect to the functional parametrizing the approximation. Authors develop an update rule to evolve the network parameters such that they converge to the global optimum in the functional space. The method is tested in a simple inference problem, and results show that the method converges close to the exact posterior.

Some weaknesses:
- The experiments are rather limited. You have this 1d inference problem, and the discussion regarding the results is somewhat limited. How good are the NLL values? Your method does not converge to the exact posterior, so some context on how severe this gap is would be beneficial for the reader.
- Moreover, since you result is that the VI converges to the global optimum, I'm not sure if comparison against exact posterior is meaningful. Except if the exact posterior belongs to the family of the variational approximation.

---

### Official Review · Reviewer_rc8b · 2024-04-25
**Response by reviewer.**

**Rating:** 6
**Confidence:** 2

**Review:**

## Summary

This study demonstrates the existence of a global solution for variational inference (VI) using the training objective of expected forward KL divergence under mild assumptions. Additionally, this work investigates the convergence analysis to a global solution when employing gradient descent to solve the optimization problem. In this procedure, the authors employ the gradient flow using the neural tangent kernel (NTK).



-------------------
## Pros

* Introducing the condition for achieving the global solution in variational inference (VI) appears to be a novel contribution, as existing VI methods primarily concentrate on the training objective of reverse KL divergence, which has been known as non-convex optimization problem.

*  Furthermore, presenting the convergence analysis for achieving the global solution also appears to be a novel contribution.



-------------------

##  Suggestions.

To the best of my knowledge, VI using the forward KL divergence as the training objective generally results in overdispersed variational posteriors, as mentioned by the author. I believe that this property is less favorable compared to the reverse KL divergence, which is commonly used for training Bayesian Neural Networks (BNNs) or generative models, as it can easily lead to the trained model being underfitted to the dataset.

* Introducing additional discussion on the merits of using the forward KL divergence as the training objective in either the "introduction" or "experiment sections" would enhance the value of this work's results in the revised manuscript.

---

### Official Review · Reviewer_wAMF · 2024-04-25
**Strong paper that provides important analyses for amortized variational inference with the forward KL**

**Rating:** 8
**Confidence:** 4

**Review:**

The paper studies convergence of variational inference with exponential family distributions using the expected forward KL divergence where the expectation is over the marginal distribution of the generating process. They prove that when the exponential family parameters are outputs of some function then the divergence is convex in the function. They then show that for a single hidden layer scaled ReLU fully-connected neural network that as the parameters are evolved according to continuous gradient descent dynamics the learned function converges to the global function solution in the infinite layer width limit. Their experimental results confirm that as the layer width is increased the neural network's final negative log-likelihood matches that of the neural network linearized about its initial parameters which is the infinite width limiting behaviour of the neural network.

The paper was very clear to read and easy to understand the main contributions. They provides key analyses of a forward KL objective that isn't typically used in variational inference and thus will be of great interest to those seeking to use this new objective as opposed to the standard reverse KL. The surprising result that continuous parametric gradient descent in the infinite width limit approaches the global function optimizer $f^*$ is a strong theoretical result that has practical implications in that increasing the layer width produces optimal functions that better approximate $f^*$. The experiments also provide strong evidence for this.

One confusion I had concerning the paper is the discussion in section 2.1 that their method will not work with an empirical dataset $\mathcal{D}$ because we would not be able to obtain unbiased gradient estimates. It was not clear as to why this is the case because $\mathcal{D}$ can simply be cast as a specific type of $P(X)$ and so that setting would fit precisely in the formulation for the expected forward KL that they consider.

Overall, the paper proves important results for amortized variational inference with the forward KL and I believe it will be of significant interest to the AABI community.

---

### Meta-Review · Area_Chair_iDTW · 2024-05-12

**Recommendation:** Accept (Poster)
**Confidence:** 4

**Metareview:**

This paper analyzes the forward-KL objective for variational inference. The authors show the strict convexity of the objective in the function space under certain conditions. Moreover, using the machinery of the NTK, the authors also show the strict convexity of the objective in the parameter space, when the network is overparametrized enough. This implies that under those conditions, forward-KL variational inference converges to a global optimum.

Most reviewers agree that the paper is interesting. However, one reviewer (rightly) mentioned that the paper could be improved more: (i) the title is a bit misleading---the authors only study the forward-KL VI, not the more popular backward-KL one, (ii) the paper (softly) violates the AABI format---there is no abstract in the pdf.

I recommend acceptance but the authors should take the reviewers' comments seriously in preparing the final version.

---

### Decision · Program_Chairs · 2024-05-27

Accept